# Floquet Chern insulators of light

Li He[1], Zachariah Addison[1], Jicheng Jin[1], Eugene J. Mele[1], Steven G. Johnson[2] & Bo Zhen[1]

Achieving topologically-protected robust transport in optical systems has recently been of great interest. Most studied topological photonic structures can be understood by solving the eigenvalue problem of Maxwell's equations for static linear systems. Here, we extend topological phases into dynamically driven systems and achieve a Floquet Chern insulator of light in nonlinear photonic crystals (PhCs). Specifically, we start by presenting the Floquet eigenvalue problem in driven two-dimensional PhCs. We then define topological invariant associated with Floquet bands, and show that topological band gaps with non-zero Chern number can be opened by breaking time-reversal symmetry through the driving field. Finally, we numerically demonstrate the existence of chiral edge states at the interfaces between a Floquet Chern insulator and normal insulators, where the transport is non-reciprocal and uni-directional. Our work paves the way to further exploring topological phases in driven optical systems and their optoelectronic applications.

[1] Department of Physics and Astronomy, University of Pennsylvania, Philadelphia, PA 19104, USA. [2] Department of Mathematics, Massachusetts Institute of Technology, Cambridge, MA 02139, USA. Correspondence and requests for materials should be addressed to B.Z. (email: bozhen@sas.upenn.edu)

The field of topological photonics seeks to classify and demonstrate various topological phases in Maxwell's equations, and to apply their associated robust states in optical systems[1–3]. Though initially inspired by progress in electronic systems, topological photonics has recently developed in multiple directions using its unique ingredients, such as the easy incorporation of non-Hermiticity via material gain[4–6] or radiative loss[7]. Many important applications of topological photonics, such as optical isolators and circulators, are non-reciprocal in nature, which means they are exclusive for topological phases in systems with broken time-reversal symmetry. In static structures, such topological phases are often achieved by starting with engineered degeneracies between two bands of a PhC—in an either linear (Dirac) or quadratic fashion—followed by a static perturbation that breaks reciprocity, such as gyromagnetic effects[8–11]. The resulting systems are often referred to as Chern insulators, as their topological gaps can support uni-directional modes, whose transport is protected by the topological invariant of Chern numbers. Another important method to break reciprocity is through temporal modulation[12], yet the understanding of topological phases in dynamically driven optical systems is often limited to tight-binding models of coupled resonators[13–15] or waveguides[16–18].

Here, we study Floquet topological phases in general nonlinear PhCs under external drive and show how non-reciprocal transport can be achieved in a Floquet Chern insulator. We start by formulating the Floquet eigenvalue problem of Maxwell's equations, and show it is necessarily non-Hermitian but with real eigenvalues in many cases. After elucidating what time-reversal symmetry (T) entails in driven systems, we engineer the external drive to break T and to close and re-open Floquet gaps to change bands Chern numbers. Finally, through numerical simulations of realistic designs, we present an explicit example of a Floquet Chern insulator, along with the dispersions and locations of uni-directional chiral edge states at its interfaces with normal insulators.

## Results

**Floquet gaps and Floquet eigenvalue problem**. We start by showing that new bandgaps—Floquet gaps—can be created in driven nonlinear PhCs, which do not exist in the static band structure. We consider a two-dimensional PhC that involves second-order optical nonlinear materials such as $LiNbO_3$. The static band structure is schematically shown in Fig. 1b, and we focus on two isolated bands: $|1\rangle$ in blue and $|2\rangle$ in red, which are separated by a gap in the spectrum. When an external driving

field at frequency $\Omega$ is applied along the normal direction, the discrete spatial translation symmetry of the system is preserved, but the continuous temporal translation symmetry is broken, leaving only a discrete temporal translation symmetry. Accordingly, each band creates copies of itself—Floquet bands—shifted up or down in the spectrum by $m\Omega$, where $m$ is an integer. When $\Omega$ is slightly larger than the static gap, two of the Floquet bands, $|1, m = 0\rangle$ and $|2, m = -1\rangle$, cross, and the coupling between them $V_{21}$ opens a new gap—Floquet gap—that is controlled by the driving field. When the driving field is weak, the size of the Floquet gap is linearly proportional to the coupling strength $|V_{21}|$, meaning this gap can only be closed at momentum (**k**) points where the complex coupling term vanishes: $V_{21}(\mathbf{k}) = 0$. We later show these singular points represent the topological phase transitions between Floquet Chern insulators and normal insulators.

Next, we present the Floquet eigenvalue problem of Maxwell's equations in this system. The result (Eqs. (1a) and (1b)) is achieved by adding time-dependent nonlinear permittivity tensor $\bar{\bar{\epsilon}}_{nl}(t)$—determined by both the nonlinear material and the driving field—into the static eigenvalue problem[1].

$$A\Psi = i\partial_t[(B_0 + B_{nl})\Psi] \tag{1a}$$

$$A = \begin{pmatrix} 0 & i\nabla\times \\ -i\nabla\times & 0 \end{pmatrix}, B_0 = \begin{pmatrix} \bar{\bar{\epsilon}}_1 & 0 \\ 0 & \mu_0 \end{pmatrix}, B_{nl} = \begin{pmatrix} \bar{\bar{\epsilon}}_{nl}(t) & 0 \\ 0 & 0 \end{pmatrix} \tag{1b}$$

where $\bar{\bar{\epsilon}}_1$ is the linear permittivity tensor, and $\Psi(t) = (\mathbf{E}, \mathbf{H})^T$ are the complex electromagnetic fields. Here, we focus on instantaneous nonlinear processes and assume all materials involved are dispersion-less and loss-less for simplicity, although dispersive medium can potentially also be included[19]. Compared to the Floquet eigenvalue problem of Schrödinger equation[20–22], our problem is different in a few unique ways. First, it is necessarily non-Hermitian, as the $i\partial_t$ term cannot commute with the $B_{nl}(t)$ term on the right hand side of Eq. (1a), though each individual term is Hermitian. Second, interestingly, the Floquet eigenvalues can be guaranteed as real under some conditions discussed later. We solve the Floquet states $\Psi(t)$, which are the eigenstates of this linear eigenvalue problem, by expanding them in the Floquet basis $|j, m\rangle = |j\rangle e^{im\Omega t}$ as $\Psi(t) = e^{i\mathbf{k}\cdot\mathbf{r} - i\epsilon t} \sum_{jm} c_{jm}|j, m\rangle$. Here, $\epsilon$ is the quasi-energy; $|j\rangle$ satisfies the static eigenvalue problem of $e^{-i\mathbf{k}\cdot\mathbf{r}} A e^{i\mathbf{k}\cdot\mathbf{r}}|j\rangle = \omega_j B_0|j\rangle$ and therefore forms a complete basis of spatial modes. Similar approach has also been used in solving the Floquet eigenstates of periodic paraxial equations[23,24]. The detailed solution is presented in Supplementary Note 1 with discussions in Note 2.

**Topological defects in momentum space**. To better illustrate some of the key concepts, we focus on an example when two bands become close to each other under driving $\omega_2 - \omega_1 \approx \Omega$, while both are far away from other bands. Hence, we restrict the trial solutions to the subspace spanned by the two bands for simplicity; however, the presented formalism is general and not limited to the two-band model. Under a further rotating-wave approximation, the Floquet eigenvalue problem can be simplified into:

$$\begin{pmatrix} \omega_2 - \Omega & -\Omega V_{21} \\ 0 & \omega_1 \end{pmatrix} \begin{pmatrix} c_{2,-1} \\ c_{1,0} \end{pmatrix} = \epsilon \begin{pmatrix} 1 & V_{21} \\ V_{21}^* & 1 \end{pmatrix} \begin{pmatrix} c_{2,-1} \\ c_{1,0} \end{pmatrix}. \tag{2}$$

As shown, this generalized eigenvalue problem is indeed non-Hermitian, but its eigenvalues can be guaranteed as real under some conditions. For example, when the driving field is exactly

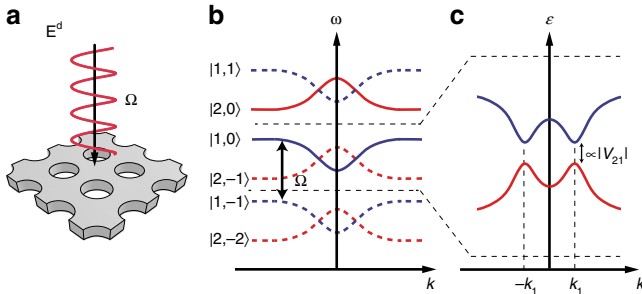

**Fig. 1** Floquet bands and gaps in a periodically driven nonlinear photonic crystal. **a** Schematic of a nonlinear photonic crystal (PhC) placed in a monochromatic driving field $\mathbf{E}^d$ at frequency $\Omega$. **b** Due to the periodic drive, static bands of the PhC (solid lines) create copies of themselves—Floquet bands (dashed lines)—by shifting up or down in the spectrum. **c** Two of the Floquet bands $|1, m = 0\rangle$ and $|2, m = -1\rangle$ cross at $\pm k_1$. Their coupling term $V_{21}$ opens a new gap $E_g$—Floquet gap—and its size is linearly proportional to the magnitude of their coupling strength $|V_{21}|$ under weak drive

on-resonance, namely $\omega_2 = \omega_1 + \Omega$, the two Floquet eigenvalues can be further simplified as: $\varepsilon_\pm \approx \omega_1 \pm 2|V_{21}|\sqrt{\omega_1 \omega_2}$. The normalized gap size is linearly proportional to $|V_{21}|$, whose magnitude is determined by both the modal overlap and the driving field strength. Furthermore, we note that both eigenvalues are necessarily real as long as we are coupling bands both at positive (or negative) frequencies ($\omega_1 \omega_2 > 0$). Physically, these scenarios are analogous to the depletable sum-frequency generation: power oscillates between a depletable pump $\omega_1$ and the sum-frequency beam $\omega_2$, but their total photon number remains fixed in time[25,26]. On the other hand, complex eigenvalues may appear when a positive-frequency mode is coupled to a negative-frequency mode ($\omega_1 \omega_2 < 0$) and the resulting Floquet modes may grow exponentially in time. These scenarios are analogous to optical parametric amplification where a non-depletable pump beam ($|\omega_1| + |\omega_2|$) amplifies the signal and idler beams[26]. In this Letter, we focus on the first situation where Floquet eigenvalues are real. Topological phase transitions can only happen at **k** points where the gap is closed, requiring the coupling term $V_{21} = 0$. This is equivalent to requiring the complex phase $\arg V_{21}$ to be undefined, or to be a topological defect[27], in **k** space. The topological phase transitions, being topological defects, are thus robust against any perturbations that modify the complex coupling terms $V_{21}$, as such perturbations cannot get rid of the topological phase transitions but shift their positions in the 3D parameter space of ($k_x, k_y, \Omega$).

Next, we show how such topological defects can be synthesized by engineering the polarization of the driving field. Our considered PhC sample is shown in Fig. 2a, which is consisted of a hexagonal lattice, with lattice constant $a$, of regions made of silicon ($\epsilon = 12.25$) and regions made of $z$-cut LiNbO$_3$ ($\epsilon_{xx} = \epsilon_{yy} = 4.97$, $\epsilon_{zz} = 4.67$). Both inversion and rotation symmetries are broken to lift all degeneracies at high-symmetry **k** points. The static band structure is calculated using Finite Element Methods (see Methods section for details) and shown in Fig. 2b. In the static structure, TE bands ($H_z, E_x, E_y$; red) are decoupled from the TM bands ($E_z, H_x, H_y$; blue), due to the mirror symmetry in the $z$ direction. However, under a driving field polarized in the $xy$ plane, TE and TM bands are coupled: specifically, the external field $E_{x,y}^d$ drives the second-order optical nonlinearity of LiNbO$_3$, $\chi^{(2)}_{zxx}$ and $\chi^{(2)}_{zyy}$, and creates $\epsilon_{xz,zx}$ and $\epsilon_{yz,zy}$ terms in the effective permittivity tensor of LiNbO$_3$. These four terms break the mirror symmetry in $z$ and couple the $E_z$ component of a TM mode to the $E_{x,y}$ components of a TE mode. By analyzing the nonlinear optical property of LiNbO$_3$, one can show only TE-TM bands are coupled via modulation in this setup, while the Floquet TE-TE or TM-TM bands will not couple to each other (see Methods for details).

We found that time-reversal symmetry ($T$) in the Floquet eigenvalue problem is defined as $V_{21}(\mathbf{k}) = V_{21}^\star(-\mathbf{k})$. Furthermore, we found $T$ is preserved when the driving field is linearly polarized and no topological Floquet gap can be opened. On the other hand, elliptically polarized driving fields break $T$. The condition on $T$ in these two scenarios can be intuitively understood by analyzing the temporal evolution of the instantaneous optical principle axes of LiNbO$_3$: under a linearly polarized monochromatic drive, one optical axis remains static, while the other two oscillate in a time-reversal symmetric manner. In comparison, under an elliptically polarized drive, all three optical axes rotate around the $z$ axis at the driving frequency and this spinning behavior breaks $T$. Detailed derivation is presented in Supplementary Notes 4 and 5.

The properties associated with time-reversal symmetry are confirmed in our simulation results of the modal coupling terms $V_{21}$ as shown in Fig. 2c, d. Specifically, under a linearly polarized

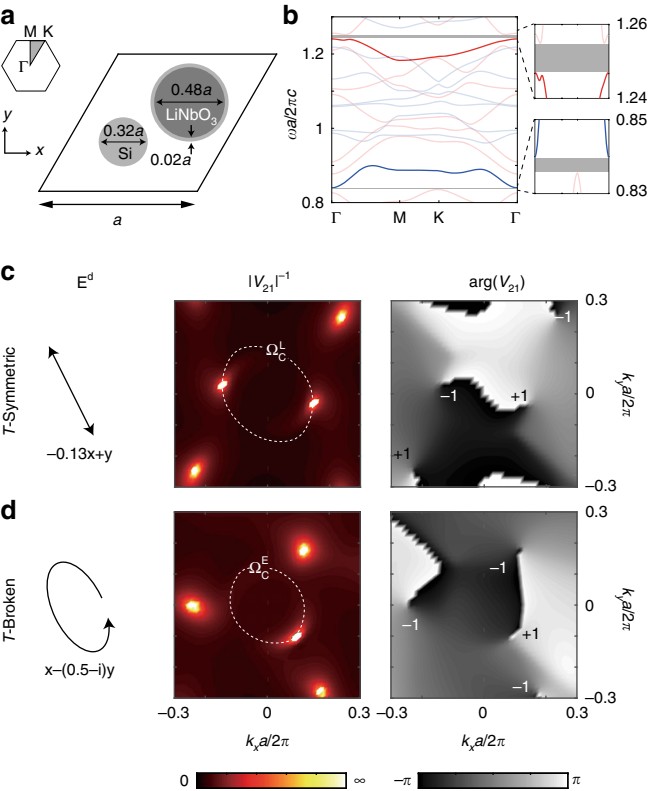

**Fig. 2** Topological charges in modal coupling terms and the influence of time-reversal symmetry. **a** Nonlinear PhC unit cell involving Si and $z$-cut LiNbO$_3$. The centers of the Si and LiNbO$_3$ rods are at $(0.55a\hat{\mathbf{x}} + a/2\sqrt{3}\hat{\mathbf{y}})$ and $(a\hat{\mathbf{x}} + a/\sqrt{3}\hat{\mathbf{y}})$ with respect to the bottom-left corner of the unit cell. **b** Static modes are separated into TE (red) and TM (blue) bands. Under a driving field polarized in the $xy$ plane, a TE band is coupled to a TM band through $\chi^{(2)}$ of LiNbO$_3$. Their coupling term $V_{21}$ is controlled by the polarization of the drive. **c** Under a linearly polarized drive ($-0.13\hat{\mathbf{x}} + \hat{\mathbf{y}}$), pairs of vortices with opposite topological charges ($\pm 1$) are found in the complex phase $\arg V_{21}$, located at opposite **k** points. At these **k** points, the modal coupling term vanishes and $1/|V_{21}| \to \infty$. **d** Under an elliptically polarized drive, $\hat{\mathbf{x}} - (0.5 - i)\hat{\mathbf{y}}$, vortices in $\arg V_{21}$ appear without any symmetry. Topological phase transition is achieved between a Floquet normal insulator and a Floquet Chern insulator through a single topological charge at $\Omega_C^E$ (dashed circle)

drive ($T$-symmetric), $V_{21}$ reduces to 0 at pairs of opposite **k** points that are related by $T$-symmetry, shown as bright spots in Fig. 2c. Furthermore, each pair of topological defects carry opposite topological charges $q$, which are defined through the winding numbers of the complex phase:

$$q = \frac{1}{2\pi} \oint_C d\mathbf{k} \cdot \nabla_\mathbf{k} \arg V_{21}. \qquad (3)$$

Here $C$ is a closed path in **k** space that encircles the defect in the counter-clockwise direction. Consequently, the Floquet gap can be closed and re-opened by tuning the driving frequency through a critical value $\Omega_C^L a/2\pi c = 0.375$ (dashed circle); however, the transitions always happen at a pair of opposite **k** points and the Floquet bands are always topologically trivial. See Supplementary Note 6 for the definition of Berry curvature and Chern number of Floquet bands. On the other hand, under an elliptically polarized drive ($T$-broken), topological defects appear without any symmetry (Fig. 2d). As a result, the Floquet gap can close and re-open at a single **k** point, as $V_{21}(\mathbf{k})$ is no longer related to

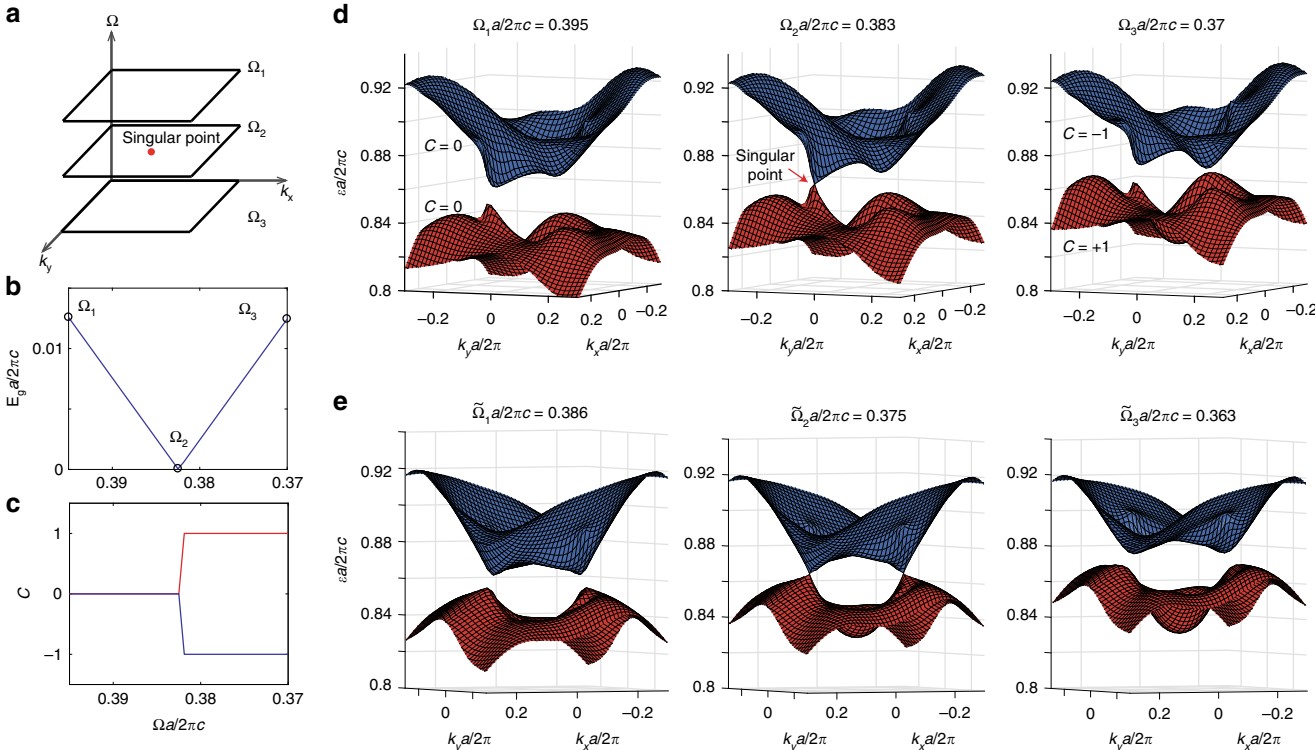

**Fig. 3** Topological phase transition points in 3D synthetic parameter space. **a** The topological phase transition point is defined in the parameter space of $(k_x, k_y, \Omega)$. **b** Under an elliptically polarized drive, the Floquet band gap closes at $\Omega_2 a/2\pi c = 0.383$, and increases linearly in its vicinity on both sides. Here we plot the partial gap size defined as the minimum of partial gap in the **k** space. **c** Under an elliptically polarized drive, the Chern numbers of the top (blue) and bottom band (red) change by 1 as $\Omega$ scans through $\Omega_2$. **d** The Floquet band structure shows a linear touching between the top and bottom bands at a singular **k** point at $\Omega_2$ (middle panel). This singular point represents a topological phase transition between a Floquet normal insulator ($\Omega_1$, left) and a Floquet Chern insulator ($\Omega_3$, right). **e** In contrast, the Floquet band gap closes and re-opens at a pair of opposite **k** points for $T$-symmetric drive when tuning driving frequency from $\tilde{\Omega}_1$ to $\tilde{\Omega}_3$ and is thus topologically trivial

$V_{21}(-\mathbf{k})$. In our system, this topological phase transition happens at another critical value $\Omega_C^E a/2\pi c = 0.381$ (dashed circle).

**Topological phase transition through unpaired topological defects.** Next, we study topological phase transitions between Floquet Chern insulators and normal insulators and show these transition points are singular points in the parameter space of $(k_x, k_y, \Omega)$ as shown in Fig. 3a. First, the Floquet band gap closes at the transition point, but grows linearly as $\Omega$ deviates from $\Omega_2$ (Fig. 3b). Furthermore, we compare the Floquet spectra near the transition point: the bands are gapped when either $\Omega > \Omega_2$ (left panel of Fig. 3d) or $\Omega < \Omega_2$ (right); however, the two Floquet bands touch at a singular point in **k** space in a linear fashion when $\Omega = \Omega_2$ (middle). We note the small difference between $\Omega_2$ and $\Omega_C^E$ arises from the difference between full Floquet formulation we adopt here and results under rotating-wave approximation. The gap size grows linearly as the system parameter deviates from a single point in the three-dimensional parameter space of $(k_x, k_y, \Omega)$, therefore, the transition points can also be interpreted as synthetic Weyl points[28–31]. We further track the Chern numbers of the Floquet bands as $\Omega$ is varied: the Chern number of the top (bottom) band changes by $-1$ (1) as the modulation frequency reduced from $\Omega_1 a/2\pi c = 0.395$ (Floquet normal insulator) to $\Omega_3 a/2\pi c = 0.37$ (Floquet Chern insulator), through $\Omega_2 a/2\pi c = 0.383$ (Fig. 3c). In addition, the Chern numbers of the two bands jump in opposite directions with their sum fixed at 0, which confirms our system is a Chern insulator. Similarly, the Floquet gap can also be closed and re-opened under linearly polarized driving fields. For example, by tuning $\Omega$

through a critical value of $\tilde{\Omega}_2 a/2\pi c = 0.375$, the Floquet gap is closed and re-opened, but at a pair of opposite **k** points. Through this process, all bands remain topologically trivial with zero Chern numbers due to the presence of $T$-symmetry.

**Chiral edge states induced by driving field.** Finally, we show the existence of chiral edge states at the interfaces between a Floquet Chern insulator (gray region in Fig. 4a) and normal insulators (white region). In this super-cell geometry, we apply periodic boundary conditions in both $x$ and $y$ directions, and these two insulators have two interfaces, top and bottom. The topological region shares the same setup as the right panel of Fig. 3d; the trivial region is driven at the same frequency $\Omega_3 a/2\pi c = 0.37$, but with a linearly polarized light $(\hat{\mathbf{x}} + 0.3\hat{\mathbf{y}})$ that preserves $T$. Through a super-cell calculation (Supplementary Note 7), all bands in the system are computed. Aside from the bulk bands in the trivial and nontrivial regions, we see chiral edge states (red and blue lines) emerge at the two interfaces with frequencies going across the topological band gap. Their mode profiles further confirm these are indeed edge states localized at the top (red) and bottom (blue) interfaces (Fig. 4c); in comparison, a bulk mode (black) is delocalized along the $y$-direction. As a control experiment, when the driving frequency is changed to $\Omega_1$ such that all regions are topologically trivial, no gapless chiral edge state is observed in such scenario (Fig. 4d). This confirms the number of chiral edge states and their traveling directions are consistent with the Floquet topological band theory results for electronic systems[32]. We note that the photon number is conserved in edge state transport. This is to be distinguished from a

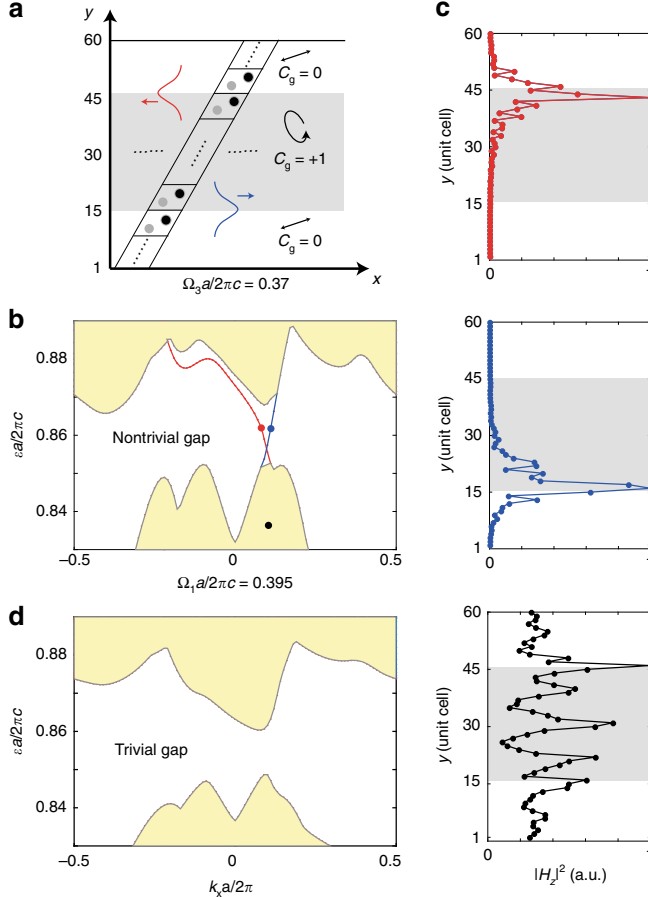

**Fig. 4** Characterizations of chiral edge states at the interfaces between a Floquet Chern insulator and normal insulators. **a** Schematic of the super-cell geometry with a Floquet Chern insulator placed in between two Floquet normal insulators. The Floquet Chern insulator is the same setup as the right panel of Fig. 3d. All regions are driven at the same frequency $\Omega_3$, and the associated gap Chern number for the Floquet insulator is $C_g = 1$. Further details of the super-cell setup can be found in Supplementary Note 7. **b** The dispersion along $k_x$ axis shows two types of modes: bulk bands in the normal and Chern insulator regions (yellow) and uni-directional chiral edge states at the top and bottom interfaces (red and blue). **c** Chiral edge states are absent in the projected dispersion when all regions are driven at frequency $\Omega_1$ before the topological phase transition. **d** Comparison among the mode profiles of a bulk state (black), the chiral edge states localized at the top interface traveling to the left (red) and at the bottom interface traveling to the right (blue)

previous study using nonlinear parametric driving by Peano et al.[33], where the photon number is not conserved, and the edge state transport becomes inelastic. Although helical spatial modulation of waveguide arrays achieves Floquet Chern insulators in the transverse plane[16], our approach breaks reciprocity for the system as a whole and thus enables optical isolation through the chiral edge states.

## Discussion

To sum up, we present a general framework to achieve Floquet topological phases in nonlinear photonic crystals, defined by the Floquet eigenvalue problems in Maxwell's equations. We show that Floquet band gaps can be closed and re-opened in a virtually arbitrary fashion by engineering the driving field (polarization and frequency). Using this framework, we propose and numerically

demonstrate a Floquet Chern insulator of light by breaking time-reversal symmetry using elliptically polarized driving fields. We show the Floquet topological phase transitions are through singular points of modal coupling terms in 3D parameter space. Finally, we numerically demonstrate the existence of chiral edge states at the interfaces between topologically trivial and non-trivial regions. Our work paves the way to further classifying and realizing topological phases in dynamically driven optical systems and their optoelectronic applications in communication and signal routing. Our method of inducing Floquet topological phases is also applicable to other wave systems, such as phonons, excitons, and polaritons.

Note added: During the completion of this work, we became aware of a related study by Fang and Wang[34].

## Methods

**Numerical simulation of Maxwell equation using Finite Element Methods**. The band structures and mode profiles are calculated using Finite Element Methods in COMSOL Multiphysics 5.3a. Specifically, we first compute the static band structures and mode profiles using the linear permittivity in a 2D geometry with periodic boundary conditions. The modal overlaps $V_{21}(\mathbf{k})$ are calculated by taking the inner product between the two modes mediated by external drive and nonlinear susceptibility of the LiNbO$_3$. Finally, we input these coupling terms into the master equation (Supplementary Eq. (4)) to calculate the eigenvalues, mode profiles, Berry curvature, and Chern numbers of the Floquet bands.

**Band coupling via the second-order optical nonlinearity of LiNbO$_3$**. Under an driving field polarized in the $xy$ plane, TE-TM bands are coupled to each other through $\chi^{(2)}_{zxx}$ ($d_{31}$) and $\chi^{(2)}_{zyy}$ ($d_{32}$) terms of LiNbO$_3$, both of which are 5 pm·V$^{-1}$[35]. On the other hand, the $E_z$ components of TM modes cannot couple to each other via modulation, because the relevant terms, $\chi^{(2)}_{zxz}$ and $\chi^{(2)}_{zyz}$, are both 0 in LiNbO$_3$. Similarly, the $E_{x,y}$ components of TE modes cannot couple via modulation either. The resulted Floquet gap size due to band coupling is linearly proportional to both the $\chi^{(2)}$ coefficients and the driving field strength. We present the estimation on the Floquet gap size that can be possibly achieved in realistic nonlinear materials in Supplementary Note 8.

## Data availability

The data that support the findings of this study are available from the corresponding author upon reasonable request.

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

## Acknowledgements

We thank H. Zhou for discussions. L.H. was supported by NSF through the University of Pennsylvania Materials Research Science and Engineering Center DMR-1720530 and grant DMR-1838412. Work by Z.A. and E.J.M. interpreting the topological character of Floquet states was supported by DOE Office of Basic Energy Sciences under grant DE FG 02 ER84-45118. S.G.J. was supported by U.S. Army Research Office through the Institute for Soldier Nanotechnologies (W911NF-13-D-0001). B.Z. was supported by the Air Force Office of Scientific Research under award number FA9550-18-1-0133.

## Author contributions

L.H. and B.Z. conceived the idea. L.H. carried out numerical simulations. L.H., Z.A., J.J., E.M., S.J., and B.Z. discussed and interpreted the results. L.H and B.Z. wrote the papert with contribution from all authors. B.Z. supervised the project.

## Additional information

**Competing interests:** The authors declare no competing interests.

