## [Peer Review File · Nature Communications]

Reviewers' comments:

Reviewer #1 (Remarks to the Author):

Major claims:

This manuscript studies a two-dimensional photonic crystal with permittivity periodically modulated in time. The considered periodic modulation can be created by building the crystal using materials with $\chi^{(2)}$ nonlinearity and applying an external pump. When the pump breaks time-reversal symmetry, e.g. if it is elliptically polarized, it can induce a topologically non-trivial Chern insulator phase hosting one-way edge modes at interfaces with topologically trivial photonic crystals.

Novelty and interest:

The main novelty of this paper is to synthesize and extend existing ideas to a new material platform: externally-pumped nonlinear photonic crystals. The bulk of previous studies on this topic were limited to the tight binding regime of weakly coupled resonators and so far there were no successful experiments. This work will broaden interest in these ideas among experimentalists and potentially enable their realization using photonic crystals.

As presently written, the manuscript does not adequately link to prior work on these topics. Additional references and discussion would help better put this work in context and make it more useful to non-specialists.

The method of computing Floquet eigenstates of continuous photonic systems based on projection onto the Bloch modes of the original unmodulated system was previously introduced in the following references (applied to the paraxial equation, in contrast to Maxwell's equations in this work):

Leykam et al, Phys. Rev. Lett. 117, 013902 (2016)

Plotnik et al, Phys. Rev. B 94, 020301(R) (2016)

The works by Peano et al [Ref. 9 and Nature Comms. 7, 10779 (2016)] seem to consider the same setup, but in the tight binding limit: a 2D photonic lattice with $\chi^{(2)}$ nonlinearity where a pump is used to break time-reversal symmetry and induce Chern edge states. However, their formalism considers a purely time-independent Bogoliubov Hamiltonian. Authors should comment on how exactly their system differs from these previous works - is it exactly the same physics, just described using a different formalism? Or is there some fundamental difference?

Creating Floquet Chern insulators using periodic non-Hermitian modulation was recently proposed by Li et al (arXiv: 1807.00913). Topological photonic crystals arising from periodic time modulation were recently discussed by Lustig et al, Optica 5, 1390 (2018), but they only considered the 1D case.

Other technical comments:

The abstract and introduction prominently stress the system is "nonlinear" in contrast to most previous works considering linear systems. This is misleading because the manuscript only deals with the solution of linear eigenvalue problems: the authors assume a-priori the external pump profile and that it remains undepleted. I think the nonlinear aspect should be down-played to avoid confusion with other recent works studying genuinely nonlinear light propagation such as soliton formation in topological photonics.

I do not think the interpretation of the phase transition as "synthetic Weyl points" is very useful and only adds unnecessary jargon. The more commonly-used nomenclature for point degeneracies of two-dimensional systems is "Dirac point". The linear detuning of the eigenstates as a function of Ω is generic for perturbed degeneracies and I do not think it is particularly remarkable.

At first glance Eqs. (S4) look like the Floquet eigenvalue problem is solved for a fixed spatial mode

j , neglecting Floquet coupling between different spatial modes. But if I understand correctly, this coupling is taken into account because V and H_0 are themselves matrices. This section would be easier to follow if the matrices were distinguished by $\hat{}$.

The Floquet basis is truncated at $m_c = 5$. In the absence of any corroborating simulations based on independent methods (e.g. FDTD), it is essential to demonstrate the convergence of this method as a function of the number of eigenstates used for the calculation. Can the authors provide some estimate of the fidelity of the calculated eigenstates?

Authors consider a 2D photonic crystal, which assumes in-plane light confinement. But the schematic in Fig. 1(a) shows a pump applied at normal incidence. Can the authors comment on how to realize such a setup while maintaining in-plane localization of the topological modes of interest? Presumably this will require sandwiching the nonlinear crystal between a cladding crystal designed to transmit at the pump frequency while simultaneously having a band-gap coinciding with the topological band-gap. This seems like an additional complication for experimentalists. Would this idea also work for an in-plane pump?

Reviewer #2 (Remarks to the Author):

In the manuscript entitled "Floquet Chern Insulators of Light," the authors describe a photonic crystal platform which, when driven by an externally imposed elliptically polarized field, realizes a photonic Floquet Chern insulator, along with a corresponding topological invariant and chiral edge states. This is a truly important work, which, if realized experimentally, may come to represent the most defining achievement of topological photonics since the early experiments in microwave and optical PCs. Therefore I unreservedly recommend publication, with a few comments that I hope the authors will find useful:

- The authors state that the problem is necessarily non-Hermitian - I wonder if Eq. (1a,b) can be represented as a Hermitian generalized eigenvalue problem, especially because the eigenvalues end up being real-valued. There are a number of wave equations that, in a certain form, appear to be non-Hermitian, but can be manipulated into a generalized Hermitian form. The absolute simplest example is a 2x2 PT-symmetric system with gain and loss on the diagonal. Although it appears to be non-Hermitian, it has a fundamentally Hermitian structure that is responsible for its real eigenvalues (below threshold). Is this the case here?

- I wonder if the picture of the Weyl point at the transition is fundamentally important. For any Chern insulator, there is a Weyl point present at the transition frequency (as a function of k_x , k_y , and some other third parameter). This is manifest by the fact that 2D Chern insulators can be thought of as living on the Brillouin zone torus with a magnetic monopole inside. That monopole is of course the Weyl point. Of course the authors' description is correct, but I feel it may be somewhat superfluous. I would recommend leaving this to the authors to consider and decide.

- I was pleased to see that the authors discussed specific experimental parameters in the supplementary information section. This will certainly help the reader to decide on the possible realizability of the experiment. In this referee's opinion, it is doable but tough... meaning that it will be done eventually.

The bottom line that this is an elegant and path-breaking work, and should most certainly be published in Nature Communications in my view.

Reviewer #1 (Remarks to the Author):

Major claims:

This manuscript studies a two-dimensional photonic crystal with permittivity periodically modulated in time. The considered periodic modulation can be created by building the crystal using materials with $\chi^{(2)}$ nonlinearity and applying an external pump. When the pump breaks time-reversal symmetry, e.g. if it is elliptically polarized, it can induce a topologically non-trivial Chern insulator phase hosting one-way edge modes at interfaces with topologically trivial photonic crystals.

Novelty and interest:

The main novelty of this paper is to synthesize and extend existing ideas to a new material platform: externally-pumped nonlinear photonic crystals. The bulk of previous studies on this topic were limited to the tight binding regime of weakly coupled resonators and so far there were no successful experiments. This work will broaden interest in these ideas among experimentalists and potentially enable their realization using photonic crystals.

Reply

We thank reviewer #1 for the positive comments on our manuscript.

Comment 1

As presently written, the manuscript does not adequately link to prior work on these topics. Additional references and discussion would help better put this work in context and make it more useful to non-specialists.

The method of computing Floquet eigenstates of continuous photonic systems based on projection onto the Bloch modes of the original unmodulated system was previously introduced in the following references (applied to the paraxial equation, in contrast to Maxwell's equations in this work):

Leykam et al, Phys. Rev. Lett. 117, 013902 (2016)

Plotnik et al, Phys. Rev. B 94, 020301(R) (2016)

Reply 1

We thank the reviewer for pointing out the two relevant papers, which we have now added on page 4:

“Similar approach has also been used in solving the Floquet eigenstates of periodic paraxial equations.”

Comment 2

The works by Peano et al [Ref. 9 and Nature Comms. 7, 10779 (2016)] seem to consider the same setup, but in the tight binding limit: a 2D photonic lattice with $\chi^{(2)}$ nonlinearity where a pump is used to break time-reversal symmetry and induce Chern edge states. However, their formalism considers a purely time-independent Bogoliubov Hamiltonian. Authors should comment on how exactly their system differs from these previous works - is it exactly the same physics, just described using a different formalism? Or is there some fundamental difference?

Reply 2

There are several fundamental differences between the two mentioned works by Peano et al (Ref. 9 and Nature Comms. 7, 10779 (2016)) and our work.

Specifically, the *static* system in Ref. 9, without driving fields, *is already topological*: the band Chern numbers and associated chiral edge states are induced by a *static* gauge field. In contrast, *our static system*, without driving fields, *is topologically trivial*, and we focus on how an external pump beam can induce a normal-to-Chern insulator transition. Therefore, these two works are on completely different subjects.

The other work by Peano *et al* (Nature Comms. 7, 10779 (2016)) studied a time-independent Bogoliubov Hamiltonian, where they used a high-frequency driving field to create signal and idler beams, or vice versa. Therefore, *the photon number is not conserved in their system*. Physically, this is an optical parametric amplification process, which, in our Floquet language, corresponds to when positive- and negative-frequency bands (ω_1, ω_2) are coupled through a driving field at Ω : $\omega_1\omega_2 < 0$, $\Omega = |\omega_1| + |\omega_2|$. As noted on page 5 of our manuscript, this regime (*without photon number conservation*) is not the focus of our work.

In contradistinction, our work focuses on scenarios when *the photon number is conserved*. These are analogous to sum-frequency generation, where bands both at positive, or negative, frequencies are coupled via drive ($\omega_1\omega_2 > 0, \Omega = |\omega_2| - |\omega_1|$). This fundamental difference between the two works is also manifested in the properties of chiral edge states: the chiral edge state transport is elastic in our study, rather than inelastic, as in the work by Peano *et al*.

To further clarify the differences between our work and these references, we have now added a sentence on page 9:

“We note that the photon number is conserved in edge state transport. This is to be distinguished from a previous study using nonlinear parametric driving by Peano et. al., where the photon number is not conserved, and the edge state transport becomes inelastic.”

Comment 3

Creating Floquet Chern insulators using periodic non-Hermitian modulation was recently proposed by Li et al (arXiv:1807.00913). Topological photonic crystals arising from periodic time modulation were recently discussed by Lustig et al, Optica 5, 1390 (2018), but they only considered the 1D case.

Reply 3

We thank the reviewer for pointing out the two relevant papers, both of which are now added on page 2. These two works study very different subjects than ours:

Li et al (arXiv:1807.00913) achieves a Floquet Chern insulator using temporal gain/loss modulations, where they solve the tight binding model of coupled resonators. In comparison, one major point of our paper is to solve the full Maxwell equations in the Floquet picture and identify topological phases within.

Lustig et al, Optica 5, 1390 (2018), studies the Zak phase associated with the 1D momentum bands (k as a function of ω) of spatially uniform samples. In contrast, we study Chern numbers associated with 2D energy bands (ω as a function of \mathbf{k}) of spatially patterned samples.

Comment 4

The abstract and introduction prominently stress the system is "nonlinear" in contrast to most previous works considering linear systems. This is misleading because the manuscript only deals with the solution of linear eigenvalue problems: the authors assume a-priori the external pump profile and that it remains undepleted. I think the nonlinear aspect should be down-played to avoid confusion with other recent works studying genuinely nonlinear light propagation such as soliton formation in topological photonics.

Reply 4

We agree with the reviewer: the Floquet topological states studied here are eigensolutions of linear eigenvalue problems. To avoid potential confusion, we have now removed all claims related to "nonlinear systems", and use "nonlinear photonic crystals" instead. Furthermore, we add one sentence on page 4:

"We solve the Floquet states $\Psi(t)$, which are the eigenstates of this linear eigenvalue problem, by expanding them..."

Comment 5

I do not think the interpretation of the phase transition as "synthetic Weyl points" is very useful and only adds unnecessary jargon. The more commonly-used nomenclature for point degeneracies of two-dimensional systems is "Dirac point". The linear detuning of the eigenstates as a function of Ω is generic for perturbed degeneracies and I do not think it is particularly remarkable.

Reply 5

We agree with the referee in the sense that: topological phase transitions in our system can also be interpreted as Dirac points in 2D \mathbf{k} -space. However, we believe our Weyl point interpretation better captures the robustness, and thus the fundamental nature, of these transition points.

Specifically, using the Weyl point interpretation, it is straightforward to show that: any perturbation of the system that modifies the complex coupling terms V_{21} (which has two degrees of freedom) will not get rid of a topological transition point. Instead, it will only shift its position in the 3D parameter space. In other words, it is always possible to recover a topological phase transition point at a nearby \mathbf{k} point under a similar driving frequency Ω . Given there is no symmetry in our Floquet system, this feature of robustness is less straightforward to explain using the 2D Dirac point picture.

Furthermore, we believe this synthetic Weyl point picture can also help readers understand topological phase transitions in Floquet systems by drawing the analogy to normal-to-Chern insulator transitions in 3D crystals - another feature that is harder to explain using the 2D Dirac point interpretation.

To better convey our points, we have now added the following sentence on page 6:

"The topological phase transitions, being synthetic Weyl points, are thus robust against any perturbations that modify the complex coupling terms V_{21} , as such perturbations cannot get rid of the topological phase transitions but shift their positions in the 3D parameter space of (k_x, k_y, Ω) ."

Comment 6

At first glance Eqs. (S4) look like the Floquet eigenvalue problem is solved for a fixed spatial mode j , neglecting Floquet coupling between different spatial modes. But if I understand correctly, this coupling is taken into account because V and H_0 are themselves matrices. This section would be easier to follow if the matrices were distinguished by $\hat{\{}}$.

Reply 6

Yes, we took the coupling between different spatial modes into account. We have now distinguished matrices by $\hat{\{}}$, as suggested by the reviewer.

Comment 7

The Floquet basis is truncated at $m_c = 5$. In the absence of any corroborating simulations based on independent methods (e.g. FDTD), it is essential to demonstrate the convergence of this method as a function of the number of eigenstates used for the calculation. Can the authors provide some estimate of the fidelity of the calculated eigenstates?

Reply 7

To address this question, we have now added a detailed discussion on the convergence of Floquet eigenstate calculations in Section I of the Supplementary Information.

To prove convergence, we first show c_{jm} is highly localized in m for a specific Floquet state (the bottom band at the Γ point in Fig.3d). As shown in SFigure 1a, the two dominant components of c_{jm} indicate that the Floquet mode arises from the hybridization between Floquet basis $|1, m = 0\rangle$ and $|2, m = -1\rangle$, while the contributions from other Fourier orders are negligible.

To further quantify the convergence of c_{jm} throughout all Floquet bands, we evaluate the worst-case scenario, defined as:

$$d(n) = \min \left(\sum_{|m| < n} \sum_{j=1,2} |c_{jm}(\mathbf{k})|^2 \right) \text{ for all } \mathbf{k} \in \text{B.Z.}$$

for the two Floquet bands in Fig.3d, respectively. The calculated $d(n)$ approaches 1 for $n \geq 2$ (SFig.1b), suggests our truncation of the Floquet basis at $m_c = 5$ is good enough for the calculation.

Figure 1: **The convergence of Floquet eigenstates calculation with truncated Floquet basis.** **a**, The localization of c_{jm} in m for a specific Floquet state (the bottom band at the Γ point in Fig.3d). **b**, The localization of $d(n)$ for two Floquet bands in Fig.3d.

Comment 8

Authors consider a 2D photonic crystal, which assumes in-plane light confinement. But the schematic in Fig. 1(a) shows a pump applied at normal incidence. Can the authors comment on how to realize such a setup while maintaining in-plane localization of the topological modes of interest? Presumably this will require sandwiching the nonlinear crystal between a cladding crystal designed to transmit at the pump frequency while simultaneously having a band-gap coinciding with the topological band-gap. This seems like an additional complication for experimentalists. Would this idea also work for an in-plane pump?

Reply 8

It is important to distinguish between the pump beam (not a resonance of the photonic crystal) and the topological bands (eigenmodes of the photonic crystal). In the proposed experiment, the topological modes of interest can be guided modes, which lie below the light line and enjoy confinement from the index contrast in the normal direction. They may also be guided mode resonances that lie above the light line, as long as their radiative loss rates are small compared to other relevant energy scales in the system. In either case, the driving field can be applied in the normal direction using the free-space plane wave.

For an in-plane pump scenario as mentioned by the referee, the finite in-plane wave vector breaks the original translational symmetry of the crystal and couples Bloch states at different crystal momenta. Such an in-plane pump can also break time reversal symmetry and achieve a Floquet Chern insulator phase. The resulting Floquet band structure can be potentially solved by expanding the Floquet basis of our current formalism.

Reviewer #2 (Remarks to the Author):

In the manuscript entitled "Floquet Chern Insulators of Light," the authors describe a photonic crystal platform which, when driven by an externally imposed elliptically polarized field, realizes a photonic Floquet Chern insulator, along with a corresponding topological invariant and chiral edge states. This is a truly important work, which, if realized experimentally, may come to represent the most defining achievement of topological photonics since the early experiments in microwave and optical PCs. Therefore I unreservedly recommend publication, with a few comments that I hope the authors will find useful:

Reply

We thank reviewer #2 for the positive comments on our manuscript.

Comment 1

The authors state that the problem is necessarily non-Hermitian - I wonder if Eq. (1a,b) can be represented as a Hermitian generalized eigenvalue problem, especially because the eigenvalues end up being real-valued. There are a number of wave equations that, in a certain form, appear to be non-

Hermitian, but can be manipulated into a generalized Hermitian form. The absolute simplest example is a 2x2 PT-symmetric system with gain and loss on the diagonal. Although it appears to be non-Hermitian, it has a fundamentally Hermitian structure that is responsible for its real eigenvalues (below threshold). Is this the case here?

Reply 1

Generically Eq. 1a,b cannot be represented as a *Hermitian definite pencil*, a special case of a *generalized Hermitian eigenvalue problem* ($A\psi = \lambda B\psi$, where A and B are Hermitian matrices and B is positive definite, λ is a real eigenvalue). This is because a Hermitian definite pencil necessarily has real eigenvalues, whereas the eigenvalues of Eq. 1a,b can be complex (see for example page 5). However, in specific cases where $\omega_1\omega_2 > 0$, the eigenvalues can be guaranteed as real and the associated eigenvalue problem can be shown as *pseudo-Hermitian*. In this limit, we are still unaware if there exists a basis transformation that convert our non-Hermitian eigenvalue problem with real spectrum into a Hermitian definite pencil.

Comment 2

I wonder if the picture of the Weyl point at the transition is fundamentally important. For any Chern insulator, there is a Weyl point present at the transition frequency (as a function of k_x , k_y , and some other third parameter). This is manifest by the fact that 2D Chern insulators can be thought of as living on the Brillouin zone torus with a magnetic monopole inside. That monopole is of course the Weyl point. Of course the authors' description is correct, but I feel it may be somewhat superfluous. I would recommend leaving this to the authors to consider and decide.

Reply 2

We agree with the referee that the topological phase transition can be understood in 2D \mathbf{k} space, where a Weyl point manifest itself as magnetic monopole inside the 2D Brillouin zone torus. However, we believe our Weyl point interpretation better captures the robustness, and thus the fundamental nature, of these transition points.

Specifically, using the Weyl point interpretation, it is straightforward to show that: any perturbation of the system that modifies the complex coupling terms V_{21} (which has two degrees of freedom) will not get rid of a topological transition point. Instead, it will only shift its position in the 3D parameter space. In other words, it is always possible to recover a topological phase transition point at a nearby \mathbf{k} point under a similar driving frequency Ω . Given there is no symmetry in our Floquet system, this feature of robustness is less straightforward to explain using the 2D Dirac point picture.

Furthermore, we believe this synthetic Weyl point picture can also help readers understand topological phase transitions in Floquet systems by drawing the analogy to normal-to-Chern insulator transitions in 3D crystals - another feature that is harder to explain using the 2D Dirac point interpretation.

To better convey our points, we have now added the following sentence on page 6:

“The topological phase transitions, being synthetic Weyl points, are thus robust against any perturbations that modify the complex coupling terms V_{21} , as such perturbations cannot get rid of the topological phase transitions but shift their positions in the 3D parameter space of (k_x, k_y, Ω) ”

Comment 3

I was pleased to see that the authors discussed specific experimental parameters in the supplementary information section. This will certainly help the reader to decide on the possible realizability of the experiment. In this referee's opinion, it is doable but tough... meaning that it will be done eventually.

The bottom line that this is an elegant and path-breaking work, and should most certainly be published in Nature Communications in my view.

Reply 3

We thank reviewer #2 for the positive comments on our manuscript.

REVIEWERS' COMMENTS:

Reviewer #1 (Remarks to the Author):

The revised manuscript satisfactorily addresses all the technical queries I raised in my previous report.

I am not convinced by the authors' response on the description of the topological transition as a synthetic Weyl point (an issue also raised by reviewer #2). The basic property of topological phases is that they are robust against small (symmetry-protecting) perturbations. Therefore it is unsurprising that small perturbations to the coupling do not completely remove the topological phase and only shift the transition point in the parameter space. I do not see any benefit in using this 3D terminology to describe their 2D system.

Prior literature on these topics raised in my previous report is now adequately cited. A minor comment: since the authors focus on a regime in which the system is stable and the photon number is conserved (making system pseudo-Hermitian), I am not sure it is necessary to stress non-Hermitian topological phases and band theory so prominently in the abstract and introduction.

A minor comment on the bottom of page 2: "show how non-reciprocal transport can be achieved in, what we call, a Floquet Chern insulator."

I do not think "what we call" should be used here, given Floquet Chern insulators were proposed theoretically in Phys. Rev. B 79, 081406(R) (2009) and Ref. [25] and observed in photonics in Ref. [19]. Otherwise this gives the misleading impression that authors are proposing and naming this Floquet Chern insulator phase for the first time, rather than merely showing how this well-known topological phase can be implemented in a photonic crystal described by Maxwell's equations.

Reviewer #2 (Remarks to the Author):

I continue to be supportive of the publication of this manuscript in Nature Communications.

Reviewer #1 (Remarks to the Author):

The revised manuscript satisfactorily addresses all the technical queries I raised in my previous report.

Reply

We thank reviewer 1 for favorable comments and constructive advice.

I am not convinced by the authors' response on the description of the topological transition as a synthetic Weyl point (an issue also raised by reviewer #2). The basic property of topological phases is that they are robust against small (symmetry-protecting) perturbations. Therefore, it is unsurprising that small perturbations to the coupling do not completely remove the topological phase and only shift the transition point in the parameter space. I do not see any benefit in using this 3D terminology to describe their 2D system.

Reply 1

We agree with the reviewer that most features of our topological phase transition can be understood using 2D Dirac point interpretations, though the 3D synthetic Weyl point interpretation is also valid.

To avoid potential confusions for general readers, we have now removed all content related to the “Weyl point” interpretation from the manuscript and the figures, with the exception of only one brief summary sentence on page 8:

“The gap size grows linearly as the system parameter deviates from a single point in the three-dimensional parameter space (k_x, k_y, Ω) , therefore, the transition point can also be interpreted as synthetic Weyl point.”

Prior literature on these topics raised in my previous report is now adequately cited. A minor comment: since the authors focus on a regime in which the system is stable and the photon number is conserved (making system pseudo-Hermitian), I am not sure it is necessary to stress non-Hermitian topological phases and band theory so prominently in the abstract and introduction.

Reply 2

We have now removed the discussion on “non-Hermitian topological phases and band theory” in the abstract and introduction.

A minor comment on the bottom of page 2: "show how non-reciprocal transport can be achieved in, what we call, a Floquet Chern insulator."

I do not think "what we call" should be used here, given Floquet Chern insulators were proposed theoretically in Phys. Rev. B 79, 081406(R) (2009) and Ref. [25] and observed in photonics in Ref. [19]. Otherwise this gives the misleading impression that authors are proposing and naming this Floquet Chern insulator phase for the first time, rather than merely showing how this well-known topological phase can be implemented in a photonic crystal described by Maxwell's equations.

Reply 3

We have now removed “what we call” in the revised manuscript as suggested by the reviewer.

Reviewer #2 (Remarks to the Author):

I continue to be supportive of the publication of this manuscript in Nature Communications.

Reply

We thank reviewer 2 for the favorable assessment and support.